# A Qualitative Study of the Pain Experiences of Children and Their Parents at a Canadian Children’s Hospital

**DOI:** 10.3390/children9121796

**Published:** 2022-11-23

**Authors:** Elise Kammerer, Joshua Eszczuk, Katie Caldwell, Jacob Dunn, Sharon Appelman-Eszczuk, Jennifer Dunn, Megan MacNeil, Samina Ali

**Affiliations:** 1Department of Pediatrics, Faculty of Medicine & Dentistry, University of Alberta, Edmonton, AB T6G 1C9, Canada; 2Women and Children’s Health Research Institute, Faculty of Medicine & Dentistry, University of Alberta, Edmonton, AB T6G 1C9, Canada; 3Faculty of Kinesiology, University of Alberta, Edmonton, AB T6G 2H9, Canada; 4Faculty of Medicine & Dentistry, University of Alberta, Edmonton, AB T6G 2B7, Canada; 5Alberta Health Services, Edmonton, AB T6G 2B7, Canada; 6School of Public Health, University of Alberta, Edmonton, AB T6G 1C9, Canada

**Keywords:** pediatric pain, pain management, advocacy, patient perspectives, family-centred care, tertiary healthcare

## Abstract

Current literature is lacking in describing families’ experiences in being involved in children’s pain management. This study sought to understand children and their parents’ experiences with pain management at a tertiary care children’s hospital. Twelve child–parent dyads were recruited to participate in the study from January to August 2022. Children and their parents chose whether to be interviewed together or separately. Transcripts were analyzed using inductive, data-driven codes. Codes and themes were developed using a codebook and member-checking. Three main themes were identified: a. Painful experiences can have a significant positive or negative effect on families’ lives and healthcare trajectories; b. There can be a mismatch between families’ expectations of pain management and how they perceive the pain was managed; c. Families feel that they must advocate for better pain care, but often feel too intimidated to do so, or worry that their concerns will be dismissed by healthcare professionals. Families want healthcare professionals to proactively manage their children’s pain, supporting the shaping of early positive memories of the child’s healthcare interactions. Healthcare providers must further recognize that poorly treated pain can significantly impact families’ lives and should both seek and be receptive to child and parent input for better pain care.

## 1. Introduction


*“Their journey with pain was arduous and eye-opening… it terrifies me as a parent. It’s terrifying.”*
-Parent, dyad 3

Hospital and ambulatory encounters often involve pain, with up to 80% of hospitalized children having at least one painful procedure every 24 h, with a mean of 6.3 painful procedures per day [1,2,3,4]. Whether a child receives care at a hospital due to a painful condition or experiences commonly performed painful procedures while in hospital, it is likely that their pain will be undertreated [5,6,7]. Less than one-third of children in Canadian hospitals have documentation confirming that they received a pain management intervention for a painful medical procedure [1]. Similarly, children presenting to emergency departments with known painful conditions receive inadequate analgesic administration. In a large study of over 20,000 children across the United States, 86% received an analgesic, while only 45% received opioids for long-bone fractures [8]. Further, in a study of almost one million American children with appendicitis, less than 60% received any analgesia, and less than 40% received opioid analgesia, for a condition which is known to cause moderate-to-severe pain [9]. This undertreatment exists despite a wide base of evidence that is available to be implemented into practice [10,11].

When a child’s pain is undertreated, more than 60% of children and up to 50% of youth develop needle fears, which can then extend into adulthood, when 20 to 30% of adults have a needle fear [12,13,14]. Studies of infants have shown that painful procedures in the intensive care unit can result in many long-term effects in infants, including persistent EEG changes reflecting pain processing changes [15,16,17,18]. High childhood pain and fear of medical procedures can even result in future avoidance of healthcare [18]. Further, caregivers who see their child in pain often experience significant distress. A study of caregivers witnessing their child receiving intravenous cannulation demonstrated a statistically significant increase in their own heart rate and anxiety which ultimately accounted for half of the variability of their child’s stress during recovery [19]. This suggests that improving parental pain experiences may positively influence a child’s experience of pain, as well. Children’s and parents’ experiences of pediatric pain in the hospital not only impact their current healthcare encounter, but future ones as well. If the healthcare visit events are recalled negatively, children can develop negative memories and associations with the healthcare setting, and are more likely to report greater pain intensity when they experience pain again [20,21].

Children want to be active in their own pain care, and parents (as well as other caregivers) emphasize the importance of open communication with and reassurance by healthcare professionals (HCPs) regarding their child’s pain care [6]. To date, studies of parent/caregiver perspectives in this area tends to survey-based, and lacking in a deeper understanding of their experiences [5,22]. For example, a systematic review of caregivers’ informational needs related to procedural pain included subjective responses about mostly information needs, but all studies except one were quantitative and lacked an understanding of caregiver expectations and motivations as they pertain to overall pain care [5]. A single-centre study of 100 children’s satisfaction with their emergency department pain management demonstrated that children’s satisfaction was correlated with pain resolution, effective child–provider communication, and the perception that their medicine worked quickly [10]. However, this study did not qualitatively explore the children’s feelings and reasons for rating their satisfaction and pain as they did. A richer understanding of what involvement in pain care means for children and their parents (henceforth referred to as families) is needed. This study sought to explore families’ experiences with pain care at a Canadian tertiary pediatric hospital by interviewing patient–parent dyads who received care at the hospital.

## 2. Materials and Methods

### 2.1. Setting

This study was conducted at the Stollery Children’s Hospital, a tertiary care pediatric hospital in Edmonton, Canada. This centre has 218 beds and is a national leader in pediatric cardiac surgery and organ transplantation [23]. The hospital has several structural mechanisms to promote adequate pain management in children, including a Pediatric Pain Management Committee, inpatient Acute Pain Service, and outpatient Chronic Pain Clinic. The interviews for this study were conducted as part of a local quality improvement initiative to improve children’s pain management at the hospital. This data therefore received formal ethics exemption from ethics approval from the University of Alberta’s Research Ethics Board.

### 2.2. Participant Recruitment

A guide for semistructured interviews was codeveloped by the study team, including a knowledge mobilization specialist (EK), pediatrician (SA), and two patient–parent dyads who have a lived experience of pain (JE, JD, SE, JD). The interviews asked participants to discuss the child’s experiences with pain at the hospital, expectations and actual treatment of pain, how involved participants were in the child’s pain care, and whether they received knowledge mobilization resources that explained how to manage the child’s pain at home (Appendix A). Participants were recruited by scanning a QR code on a poster that was posted throughout the hospital that led them to a survey to confirm their interest in being contacted for an interview. Recruitment data was collected and managed using REDCap electronic data capture tools hosted and supported by the Women and Children’s Health Research Institute at the University of Alberta. Verbal consent was obtained from the parents and children developmentally older than 7 years prior to interviews; verbal assent was obtained from children developmentally younger than 7 years as per our local research ethics board’s requirements. Interviews were then conducted with patient–parent dyads via Zoom or telephone by a Caucasian female knowledge mobilization specialist (EK) and lasted between 20 and 45 min. Children and their parents could choose whether to be interviewed together or separately. Verbatim transcripts were produced and anonymized (EK) and analyzed by EK, JE, and KC. JE and KC identify as people with lived experience of pain. Each participant was presented with a $25 electronic gift card as a token of appreciation.

### 2.3. Analysis

The coding and subsequent analysis of the data employed guidelines for inductively developing data-driven codes [24]. The original codebook was developed by EK and KC in a shared document and included inclusion and exclusion criteria for each code. Following Boyatzis [24], the initial set of codes was developed using a subsample of the data. Themes were then identified within these subsamples and were compared. EK and KC created the codes and determined their reliability through member-checking of subsamples. EK, KC, and JE then applied the code to the remaining data and held meetings to determine whether additional inductive codes were justified. Member-checking was also used at this stage of coding to determine validity among the remaining data.

## 3. Results

Twelve patient–parent dyads were recruited, with 11 of the parent participants being female. All parent participants were their child’s biological parent. Children ranged in age from 8 to 17 years, and 5 of the children had chronic (continuous or intermittent) painful conditions. Two of the children had a developmental disability. Recruitment ceased when conceptual depth was reached [25,26]. Themes developed by the study team include: a. Painful experiences can have a significant positive or negative effect on families’ lives and healthcare trajectories; b. There can be a mismatch between families’ expectations of pain management and how they perceive the pain was managed; c. Families recognize that they must advocate for better pain care, but often feel too intimidated to do so, or feel that their concerns will be dismissed by healthcare professionals.

### 3.1. Impact of Pain on Families’ Lives and Healthcare Trajectories


*“It was definitely traumatic for both my [child] and I. It’s something that will stick with us for awhile and we’re still working through it.”*
-Parent, dyad 1

Participants who experienced procedural pain or had chronic pain shared that their experiences with pain impacted their lives outside of the hospital encounter. For example, participants who experienced procedural pain tended to categorize the experience in a binary fashion, as either positive and negative, and brought these feelings to their future medical procedures. Regardless of whether participants had a positive or negative pain experience, they had strong recollections of their experience in the hospital, which they felt influenced other aspects of their lives, beyond healthcare encounters, as well. For participants affected by chronic pain, both their procedural and chronic pain experiences in the hospital and chronic pain experiences outside of the hospital significantly affected their lives by affecting the lens with which they viewed future healthcare needs, other life experiences, and opportunities.

Procedural pain management was reported as predictive of whether pain memories become positive or negative, especially when no strategies were presented about memory reframing. Participants felt that positive experiences could have lasting effects on children and parents. For example, the parent of dyad 7 spoke about multimodal pain management strategies being effective for their child:


*“They numbed it and then froze it, and then the doctor taught [child] some breathing techniques that they could use in high-stakes situations. And that really helped. They breathed together, and that seemed to calm [child].”*


The parent of dyad 7 had positive memories associated with this event, highlighting that the breathing skills their child was taught *“is a skill you can take into your life, and [they] appreciated that.”* The child noted that *“they’re caring about if you’re hurting,”* and that the HCP team’s interventions meant that they no longer were in a lot of pain.

However, even when pain management strategies were attempted by the care team, their execution sometimes created negative memories that had a lasting impact on a child and their parent. The child of dyad 1, for example, required sutures and was provided with numbing cream and intranasal anxiolytic medication. However, the child also required a lidocaine injection when they did not have complete numbing of the area. Of the experience, the parent of dyad 1 noted:


*“So, they moved on to giving [child] the freezing with the needle. [They] were still quite anxious, and the doctor and myself had to pin [them] down while they gave [them] freezing in [their] lip … it was quite traumatic for [them].”*


This experience had a lasting impression on the parent of dyad 1, who was *“surprised about how traumatic”* the experience was for their child. The child further described the situation as being *“very uncomfortable,”* and that their main memory of the experience was that they *“just remember that it hurt.”*

When children with chronic pain presented to care at the hospital, participants noted that children’s and parents’ experiences were less influenced by hospitalization, interventions, or procedural pain caused by injections or other therapies, and more likely to be influenced by the continuing presence of chronic pain during their hospitalization and their overall lives. The parent of dyad 4 explained:


*“[Child] ended up getting fired from [their] last job because of [their] health issues, because [they were] taking time off. [Child] had problems with schoolteachers … because [they] have either missed school or had a flare up on an exam day.”*


When presenting to care to help manage an acute pain flare of a chronically painful condition, the child of dyad 4 stated that the procedural aspects of pain management such as needles were no longer a significant source of stress. They noted that they *“used to”* use numbing cream prior to injections, but no longer needed to: *“It’s kind of a hassle, and so I just figured, you know… I’m big now, might as well give it a go.”* Other patient participants with chronic pain noted a similar attitude toward procedural pain—their focus was keeping pain flares minimized, and the procedural pain associated with these diagnostics of therapeutic interventions that were painful became less important over time.

While both procedural and chronic pain experiences were reported as significantly impacting families’ lives, it tended to affect them in different ways. Procedural pain experiences without instruction on how to cope with the pain seemed to be more likely to cause negative memories for children and parents, while chronic pain experiences may be more likely to create practical restrictions on their daily lives.

### 3.2. Mismatch between Expectations and Actual Experience of Pain Management


*“It was disappointing that there was no consideration… there really was no consideration for [their] pain.”*
Parent, dyad 12

Many participants, especially the parents, spoke of expecting their child’s pain to be decreased, while maybe not completely eliminated. They also spoke of expecting pain management to be prioritized by HCPs and being surprised when it was not. When this was not the case, there existed a hurtful mismatch between the expectations and actual experiences of pain management in the hospital setting. The parent of dyad 6 shared:


*“I expected they would do something to manage [their] pain. Like… I expected something. Maybe some numbing cream. [Child] might have needed surgery, so I understood they couldn’t give [them] any pills or anything. But I did expect something offered to manage [their] pain, because [they were] in a lot of pain. And I can tell you, nothing was offered.”*


The child of dyad 6 echoed their parent’s statement, saying, *“I didn’t really have any other expectations. Just that the pain would go away.”*

This mismatch existed for participants both naïve to, and experienced with, the healthcare system. The parent of dyad 10, who worked as an HCP, noted their difficulty *“trying to be a mom and not a nurse.”* They noted feeling like an outsider to the care team and felt unable to help their child unless they revealed to the care team that they were a healthcare colleague. They noted:


*“I think my expectation would have been a more structured and systematic approach or a care pathway as opposed to saying, ‘OK, you’re going to have this infusion and now we’re stopping.’ There was no step down; it was very much all or nothing … I was disappointed that it caused a lot of additional stress and trauma for both [child] and myself … seeing [them] in that state of pain and not being able to do anything to help.”*


A few participants did have experiences with congruency between expectations and actual pain experiences at the hospital. Dyad 9 shared that the emergency physician *“had a cell phone full of animal pictures and videos”* (parent, dyad 9), and the child noted that they knew *“the stitches weren’t going to hurt because I was so distracted by the pictures.”* The parent recalled feeling that the physician *“went above and beyond,”* and that distracting their child was *“incredible.”*

While some participants did indeed have positive pain care experiences in the hospital, those with more negative experiences tended to share that their expectations were not being met, even when advocating for better pain care. This mismatch caused stress and anxiety for both parents and children and created a more negative narrative when recalling the pain experience at the hospital.

### 3.3. Comfort with Advocacy


*“There’s been a couple of times where I’ve gone in… I still got help, but I don’t think they’re taking me serious[ly]. When kids come in and say, ‘This is what’s going on, I’m in a lot of pain,’ they need to take that serious[ly].”*
Child, dyad 4.

When participants experienced a mismatch between their expectations and actual management of pain at the hospital, almost all considered advocating for better pain care. There was a marked difference in willingness and comfort to do so, however, based on whether participants were experienced with, or naïve to, the healthcare system. Participants who were experienced in the healthcare system (either because of chronic illness or experience working as an HCP) were more likely to continue to advocate even when they felt dismissed by HCPs.

For dyad 4, the parent recalled a time when their child went in for a regular treatment to manage their chronic pain. Both the parent and the child needed to advocate for a change to the child’s IV to different HCPs before they were believed. They shared:


*“One nurse put in [their] IV and left to another patient, but for whatever reason, I think maybe the needle was stuck in a little bit too far, because [they’ve] had it done a gazillion times and it was hurting. Another nurse came in and [they] said it was hurting and the nurse was like, ‘Really, is it?’ like he didn’t believe [them]. So then [they] called for the other nurse across the hall and she said, ‘You know what? I might have put it in a little bit too deep… you might need to pull it out a little bit.’ And then that other nurse helped [them].”*


Some HCP parents whose advocacy was dismissed by the treating HCPs felt the need to reveal their professional to be taken more seriously. The parent of dyad 10 stated:


*“I told them I’m in healthcare … as soon as they knew, I found it was much easier for me to say, ‘It’s been 6 h since [their] last dose of Dilaudid. [They] are at a ten out of ten. [They] are screaming and suffering and need something for [their] pain.’ And there wasn’t that pushback anymore as soon as they knew.”*


In contrast, participants more naïve to the healthcare system whose children had poorly managed pain often left the encounter feeling guilty that they did not advocate more for their child. The parent of dyad 6 noted:


*“I guess I failed as a [parent] because I should’ve asked. I didn’t feel comfortable asking. I felt that if the doctors [wanted to do something]… I felt like it was their place to do something.”*


For most participants, there was an expectation that HCPs should lead the child’s pain management, and that advocacy was only needed at times where treatment or results were not matching their expectations for comfort or pain care. This approach was discussed both by those more and less familiar with the healthcare system.

## 4. Discussion

Participants in this study had largely polarized experiences with in-hospital pain management that were either very positive or very negative. All families felt that the child’s pain experience in the hospital had a significant effect on both the child and their parents that extended beyond the healthcare encounter, to their lives outside of the hospital. For participants with more negative experiences, there was often a mismatch between their expectations and the treatment received for the child’s pain. When these expectations were not met, these children and parents felt compelled to advocate, even though the fear of being dismissed by an HCP was a real concern, regardless of whether they were experienced with the healthcare setting or not. Families with positive experiences valued the education provided to them by HCPs and planned to use the learned strategies in the future.

Children who had chronic pain (and their parents) reported different pain-related needs and expectations in the hospital setting than children who underwent painful medical procedures, without having an underlying chronic pain condition. Children who have chronic pain often require multimodal interventions to help manage their longstanding pain and to improve function [27,28]. Interestingly, participants with chronic pain in this study noted that the procedural pain associated with diagnostic or therapeutic interventions such as needle pokes became less distressing over time. It seems that the immediate management of procedural pain was very important to children without chronic conditions, as the procedural pain experience, in isolation, carried more significance to them than for children who experienced chronic pain and unfortunately may experience procedural pain more frequently, as a part of their chronic pain management plan (e.g., by using injections). Attention should be paid to carefully developing a plan to manage procedural pain for all children, with consideration of their prior experiences, or lack thereof [29]. While few, if any, children desire painful medical procedures, healthcare-naïve children, lacking a constant pain-related reference point, seem to form strong early negative memories of the experience, further highlighting the important of early positive reframing for painful medical encounters. Prior research has identified that strategies such as divided attention (provision of a stimulus other than the painful one to reduce pain) or memory reframing (focusing on the positive aspects of a painful experience) may improve these memories following painful interventions [30]. All participants shared that their satisfaction with their pain and overall visit improved as pain was better managed.

Parents will often first defer to the HCP to offer a comprehensive pain management plan for their child when they come to the hospital, but still rightfully expect to be involved in the care and its development [31]. In our study, older children and those with chronic pain were also more likely to want to be directly involved in their pain care planning and execution, while younger children and those with developmental disabilities were more likely to defer to their caregiver. Interestingly, this was also a phenomenon noted in the interview process, as well, with younger participants being more likely to defer to their parent to speak about their experiences with pain in the hospital, and very rarely contradicting them. Older children’s views tended to be more nuanced, and they were more likely to contradict their parent, especially if interviewed separately from them. Future child–parent dyad research in pain should be mindful of this to best understand child perspectives outside of caregiver influence. It is important to note that this dynamic is likely a result of the interdependent relationship between child and caregivers. Still, if we truly want to understand the child perspective, we must be aware of this relationship when conducting a study and interpreting the results. HCPs must also be aware of this relationship when assessing a child’s pain, when caregivers are present.

Participants in this study shared that they expected HCPs to always be proactive in providing pain care. However, it has been previously shown that HCPs often rely on child and family advocacy before providing pain management [32], creating a mismatch between family expectations and HCP approach to pain care. It is well accepted that, after the patient themselves, parents (and other caregivers) are often most familiar with their child’s demeanour and can better identify deviations from baseline and the extent of their child’s pain and discomfort [33,34]; this would put them in an excellent position to advocate for (or with) their child. However, many participants, regardless of whether they had a positive or negative pain experience, noted that they were either too intimidated by HCPs to advocate for pain care or were dismissed when asking for additional pain management options.

## 5. Clinical Implications

Parents have indicated the following summarize themes: pain can impact children’s and caregivers’ lives outside of the hospital and can shape future interactions with care; there can be a mismatch between children’s and caregivers’ expectations of pain management and what is provided; and families have various levels of comfort advocating for better pain care and therefore expect HCPs to provide proactive pain management. For clinicians, this means HCPs should provide multimodal pain management plans that centre the child’s voice and pain experience, openly discuss and codevelop pain management plans with children and caregivers, and proactively manage children’s pain, but take advocacy seriously when a child or caregiver does advocate for better pain management (see Figure 1).

HCPs need to provide proactive, consistent, multimodal pain management strategies to children in order to improve families’ experiences and be willing to reassess and adjust the plan according to family feedback. HCPs must be aware that families may feel vulnerable when receiving care and may not be comfortable in advocating for improved pain management, even when it is needed. Again, proactive efforts must be made to increase the families’ comfort with speaking out when they need an adjustment to their pain care plan. At our local site, we will be working with patient partners, clinical champions, and administrators to make an institutional commitment to improving children’s pain care following international accreditation guidelines [35].

## 6. Limitations

Despite providing opportunities for recruitment from all areas of the hospital, there was an overrepresentation of participants who had received emergency care (6/12), although one-third of the participating families identified themselves as having chronic illness, which roughly matches its occurrence in the general pediatric population [36]. While children were provided with the opportunity to participate in the interview either at the same time as or without their parent, children who participated with their parent were less likely to provide longer answers and instead often deferred to their parent to explain their experiences. This phenomenon was enhanced when children were younger (<10 years old) or had a developmental disability. While this study originally aimed to incorporate the child’s voice, which is usually muted in both qualitative and survey studies in favour of the caregiver’s, interviewing children and parents together reinforced existing power dynamics that exist in a parent–child relationship. Similar to other qualitative studies of caregiver perspectives on pain that have been conducted, the caregiver voice was therefore more pronounced in this study [6].

## 7. Conclusions

Families want healthcare professionals to proactively manage their children’s pain, to support the shaping of early positive memories of a child’s healthcare interactions. Families want to be included in hospital pain care, but not be expected to drive or lead it. When they do speak out, families often feel intimidated or unheard. Healthcare providers should both seek and be receptive to child and parent input for better pain care.

## Figures and Tables

**Figure 1 children-09-01796-f001:**
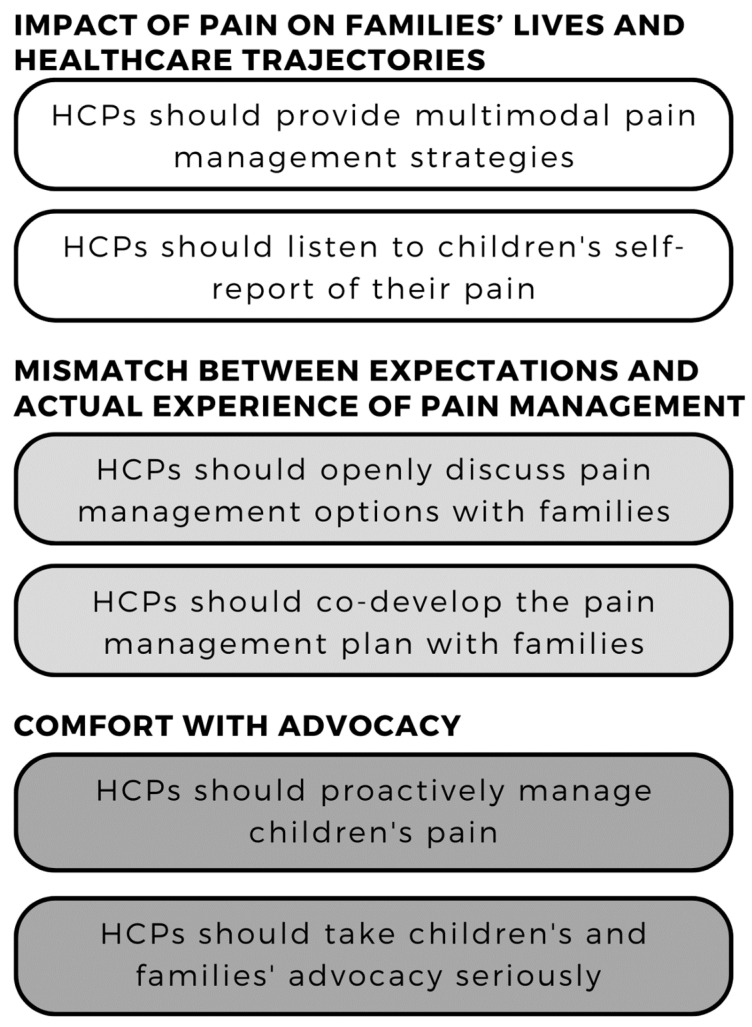
Summary of Implications for Clinical Practice.

## Data Availability

Due to the identifiable and personal nature of the content of the interviews, original data will not be made available publicly.

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
