# Peer review of "A Qualitative Study of the Pain Experiences of Children and Their Parents at a Canadian Children’s Hospital"

_children, 2022, doi:10.3390/children9121796_

Round 1

Reviewer 1 Report

Thank you for submitting the manuscript. I have read your work with great interest. The topic addressed is important and of great interest to the readers. However, I am convinced that the manuscript should undergo a thorough revision to improve its quality and readability. I believe some revisions to your manuscript are needed.

1)      I would include in the title the name of the hospital in which to have conducted the study.

2)      In the introduction, the Painless Hospital Project, born in Canada, deserves to be mentioned. In particular, pediatric references are needed to better contextualise the study. I therefore ask you to expand the introduction which must appear less discursive and more scientific. In this regard, I ask you to read and use the following references: doi: 10.1155/2008/478102. doi: 10.7417/CT.2016.1948. doi: 10.1186/s12887-022-03319-w.

3)      Eliminate sentences in italics because they weigh down the reading. I recommend that you recall them with notes in the body of the text and insert them in an appendix.

4)      In the conclusions you should include what actions you intend to take to improve pain management in your center.

5)      It is not clear to me the choice not to ask for consent for patients under 7 years of age. Explain better.

I hope my comments are useful to you.

Reviewer 2 Report

This study is important and interesting.

The sample size is small but considering the situation it can be acceptable. Besides, the authors have provided qualitative analysis.

One point to consider is if that would be possible to thematize the answers in such a way to report some cluster or themes that clinicians must be aware of in their setting to use. 

Round 2

Reviewer 1 Report

Thank you for submitting the new version of the manuscript. I am satisfied with your reviews.

Kind Regards